# Vibrational Spectroscopy of Phytochromes

**DOI:** 10.3390/biom13061007

**Published:** 2023-06-17

**Authors:** Peter Hildebrandt

**Affiliations:** Institut für Chemie, Technische Universität Berlin, Sekr. PC 14, Straße des 17. Juni 135, D-10623 Berlin, Germany; peter.hildebrandt@tu-berlin.de

**Keywords:** Raman, IR, spectroscopy, phytochrome, tetrapyrrole, photoswitch

## Abstract

Phytochromes are biological photoswitches that translate light into physiological functions. Spectroscopic techniques are essential tools for molecular research into these photoreceptors. This review is directed at summarizing how resonance Raman and IR spectroscopy contributed to an understanding of the structure, dynamics, and reaction mechanism of phytochromes, outlining the substantial experimental and theoretical challenges and describing the strategies to master them. It is shown that the potential of the various vibrational spectroscopic techniques can be most efficiently exploited using integral approaches via a combination of theoretical methods as well as other experimental techniques.

## 1. Introduction

Phytochromes constitute a class of sensory photoreceptors that utilize light as a source of information to trigger a physiological response [1,2,3,4]. They harbor a linear methine-bridged tetrapyrrole as the chromophoric unit, which, upon light absorption, undergoes a double bond isomerization around the methine bridge between rings *C* and *D* (Figure 1) [2,3]. The primary photoprocess is followed by thermal relaxations that eventually lead to functionally relevant structural changes in the protein for conversion between the parent states. These states are the red-absorbing Pr and the far-red absorbing Pfr state, which represent the physiologically inactive and active forms.

While phytochromes were initially thought to exist exclusively in plants, representatives of this photoreceptor family were later also found in bacteria and fungi [3]. Regardless of the origin, all phytochromes display the same general photo-induced reaction pattern but differ with respect to the domain composition and the type of tetrapyrrole and its binding site. Phytochromes include a tetrapyrrole-binding photosensor module composed of PAS (period/ARNT/single-minded), GAF (cGMP phosphodiesterase/adenylate cyclase/FhlA), and PHY (phytochrome-specific) domains (Figure 1A), as well as an output module with an enzymatic domain, which is frequently a histidine kinase. Plant and cyanobacterial phytochromes carry a phytochromobilin (PΦB) or phycocyanobilin (PCB), respectively, both with the Cys binding site in the GAF domain (Figure 1D). Bacterial and fungal phytochromes harbor biliverdin (BV) attached to a Cys in the PAS domain (Figure 1C) [3]. In most of the phytochromes (prototypical phytochromes), Pr is the thermodynamically stable parent state, which is also thermally recovered (dark reversion) after photo-conversion to Pfr. Only in some bacterial phytochromes are the relative thermodynamic stabilities of the parent states reversed, and Pfr is the stable dark state (bathy phytochromes) [5].

Related to phytochromes are a class of cyanobacterial photoreceptors lacking the PHY domain [6,7]. These cyanobacteriochromes (CBCR) carry a PCB chromophore, covalently attached to a Cys in the GAF domain, and also represent photoswitches between two parent states. However, these states can show absorption maxima in the entire visible spectral range from the red to the violet region [8].

**Figure 1 biomolecules-13-01007-f001:**
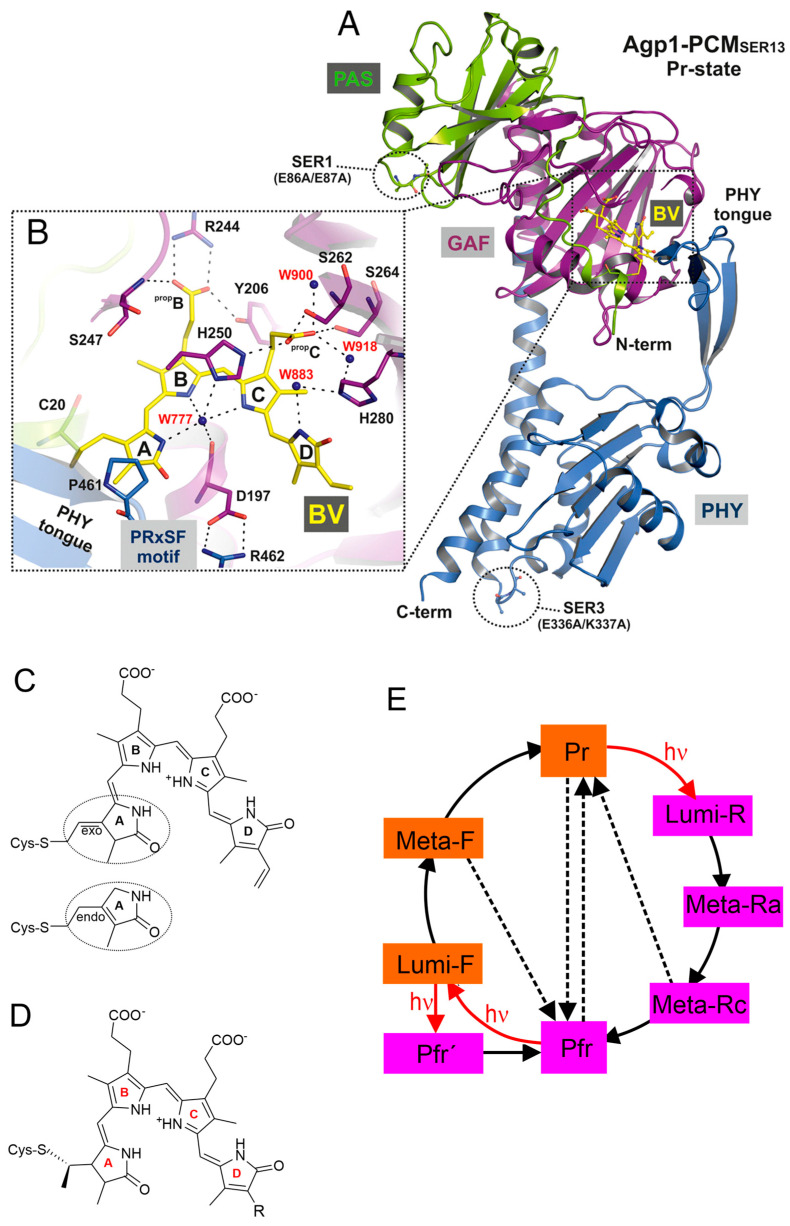
Structures and reaction scheme of phytochromes. (**A**) Structure of the photosensor core module (PCM) of the Pr state of a Agp1 variant with substitutions on the surface according to [9]. The BV chromophore and its attachment site Cys-20 are shown as balls and sticks, and carbon atoms are colored in yellow. The PAS, GAF, and PHY domains are depicted in green, purple, and blue, respectively. (**B**) Rotated close-up view (same color code as in (**A**)) of a section of the chromophore binding pocket showing the potential hydrogen bond network that links biliverdin (BV) to the protein environment, including mediating water molecules. The four pyrrole rings of BV are labeled *A* to *D*, and the propionate side chains of rings *B* and *C* are labeled prop*B* and prop*C*, respectively. ((**A**) and (**B**) were reprinted from Nagano et al. [9]). (**C**) Structural formula of BV in the *ZZZssa* configuration of Pr with endo- and exo-cyclic attachment. (**D**) Structural formula of phycocyanobilin (PCB; R = ethyl) and phytochromobilin (PΦB; R = vinyl) in the *ZZZssa* configuration of Pr. For a definition of the nomenclature, the reader should refer to the literature [10]. (**E**) Reaction scheme of the photo-induced conversion between the parent states Pr and Pfr. Red arrows indicate photoreactions; black solid and black dashed lines refer to thermal relaxations and thermal isomerization reactions, respectively. The states highlighted in orange and violet include chromophores in the *ZZZssa* and *ZZEssa* configuration, respectively.

Since the discovery of phytochromes 70 years ago [1], researchers from different disciplines strived to understand the functioning of this photoreceptor. However, in contrast to the phenomenological description of phytochrome-dependent photomorphogenesis in plants, progress in elucidating the underlying molecular processes was relatively slow in the first decades. A substantial breakthrough was achieved in the late 1990s with the discovery of phytochromes in prokaryotes and the development of efficient expression systems [11,12]. These achievements were essential for applying demanding spectroscopic techniques, such as NMR spectroscopy [13], and eventually for the successful structure determination by X-ray crystallography [14,15,16,17,18]. Among the spectroscopic techniques which were employed, infrared (IR) and resonance Raman (RR) spectroscopic techniques played a crucial role in deciphering the molecular processes in phytochromes, despite the enormous difficulties that had to be overcome in carrying out the experiments and interpreting the spectra [19,20]. With the development of hardware and novel approaches, and progress in theoretical analysis, vibrational spectroscopic methods became indispensable tools in parallel with impressive advancements in structural biology techniques [21].

This review summarizes the various approaches of vibrational spectroscopy used in studying phytochromes, their limitations and achievements, and their future potential. Instead of being a comprehensive historical survey, emphasis will be laid on the key questions of the molecular processes in phytochromes that were addressed by vibrational spectroscopies. These are specifically the photo-induced reaction mechanism and the properties of the intermediates involved (Section 3), and the coupling of chromophore and protein structural changes (Section 4). Following canonical phytochromes, Section 5 summarizes the vibrational spectroscopic studies on CBCRs that adopt different domain structures. CBCRs deserve special attention also in view of their potential application in optogenetics, an aspect that eventually leads to the last section (Section 6), which is dedicated to the contribution of vibrational spectroscopy to support the use of phytochromes in biomedicine, i.e. fluorescent phytochromes. At first, however, the RR and IR spectroscopic techniques will be briefly introduced in Section 2. For a more detailed description of experimental and theoretical background the reader is referred to the literature [20,22].

## 2. Vibrational Spectroscopic Approaches in Phytochrome Research

The first vibrational spectroscopic experiments on phytochromes were published in the late 1980s, although dedicated Raman and IR spectroscopic approaches were successfully applied to other photoreceptors much earlier [23,24,25,26]. One of the obstacles that made the vibrational spectroscopy of phytochromes an enormous challenge was the tedious isolation and purification of the protein from plants, which was, until the late 1990s, the only source for the relatively large amounts of sample needed for the experiments. Furthermore, the purified phytochrome was by far not as stable as the classical benchmark photoreceptor bacteriorhodopsin (BR) [27], and thus specifically adapted vibrational spectroscopic approaches were required.

### 2.1. IR Spectroscopic Techniques

IR absorption spectroscopy provides information about the secondary structure of proteins [21], which, in view of its large size, was not a particularly instructive method for phytochromes [28]. Thus, IR spectroscopy is mainly carried out as reaction-induced difference spectroscopy, comparing the spectra measured after irradiation with the reference spectra obtained in the dark [29,30,31]. These difference spectra exclusively display those vibrational bands of both the chromophore and the protein that undergo changes upon the reaction. Although the difference spectra between the parent states are obtained at ambient temperature, the spectral changes associated with intermediate states are measured at temperatures at which the desired state is trapped [31].

Time-resolved IR spectroscopy, like rapid-scan and step-scan spectroscopy, requires long signal accumulation times and large amounts of sample [32,33,34]. These techniques were developed on the basis of experiments with BR, which undergoes a photocycle within less than 100 ms [32,33]. Thus, within less than a second, in BR, the original photoreceptor state is recovered as a prerequisite for repetitive probing (fresh sample condition). In phytochromes, thermal recovery (dark reversion) takes much longer [3], which leads to unacceptably long measuring times. Hence, the only solution is the photo-induced recovery of the initial state, implying a demanding three-beam pump-probe-pump setup [35]. Such experiments have been successfully carried out with phytochromes covering a time range from microseconds to seconds [35,36]. Probing faster events requires a different approach based on transient absorption techniques, which allow monitoring the evolution of the IR-active vibrational bands over femtoseconds and above and can cover a wide dynamic range [37,38,39,40,41,42,43]. The experiments are based on narrow IR probe pulses, and their wavelength may be tuned over a spectral region of several hundreds of wavenumbers. In the case of monitoring localized modes, such as the C=O stretching of the chromophore, polarization-dependent measurements may even provide information about the time-dependent orientational changes of the group [42,44]. For all time-resolved IR spectroscopic techniques, a good signal-to-noise ratio is a prerequisite for a reliable spectra interpretation. In this respect, noise-reduction approaches are of particular importance, as shown, for instance, by Kübel et al. [45], who developed a generally applicable method on the basis of the time-resolved spectra of phytochromes.

Recently, two-dimensional IR spectroscopy was applied to phytochromes [46]. This technique is capable of identifying the coupling between different spectral changes and, thus, adds an additional type of information to the analysis of the structural and reaction dynamics of the photoreceptor.

### 2.2. Raman Spectroscopic Techniques

Raman spectroscopy is an important tool to selectively probe the vibrational spectrum of a chromophore upon excitation in resonance with its electronic transition (RR spectroscopy) [22,47]. In the case of phytochromes, RR spectroscopy faces the problem of fluorescence, which can obscure the RR signals, as well as uncontrolled photo-conversions. Fodor et al. were the first to present a solution by using excitation lines in the near-infrared (NIR) at 792 or 752 nm, which is shifted from the fluorescence maximum but yet sufficiently close to the first electronic transition of the tetrapyrrole chromophore (670–750 nm) to achieve good resonance enhancement [48,49]. Using a classical Raman spectrometer, these experiments suffered a bit from the low signal detection sensitivity in this spectral region. An alternative, which meanwhile has been widely applied to phytochromes, is the Fourier transform (FT) Raman spectroscopic technique, which offers the advantages of a high optical throughput and excellent frequency stability [50,51,52,53]. It is restricted to 1064-nm excitation, however, this still provides sufficient resonance enhancement of the chromophore bands such that the protein Raman bands are efficiently discriminated [51].

An interesting approach to eliminate fluorescence is shifted-excitation Raman difference spectroscopy (SERDS), in which two spectra are measured with slightly different excitation lines that are in resonance with the electronic transition of the chromophore [54]. The resultant difference spectrum cancels the fluorescence but yields a rather noisy spectrum with positive and negative Raman difference bands. These signals were simulated assuming Lorentzian band profiles and, thus, allow for calculating the absolute RR spectrum. A comparison with the spectra obtained under pre-resonance conditions demonstrated not only the reliability of SERDS but also the close similarity of the RR spectra with rigorous and pre-resonance enhancement [51,54].

The various RR spectroscopic approaches described above exploited the resonance enhancement associated with the lowest electronic transition in the red spectral region. Only a few studies have used excitation lines that were in resonance with the second transition at ca. 360–400 nm, although the fluorescence quantum yield is distinctly lower [55,56,57]. However, the high excitation energy might favor undesired side reactions.

Further techniques that circumvent fluorescence include coherent anti-Stokes (resonance) Raman spectroscopy (CARS) and surface-enhanced resonance Raman spectroscopy (SERRS). CARS probes the vibrational spectrum at the higher energy (anti-Stokes) side of the excitation line and, thus, does not interfere with fluorescence, but it is technically quite demanding [58]. SERRS can be applied when the target molecules are immobilized on nanostructured plasmonic metals like Ag or Au [59,60]. Thus, the RR scattering is enhanced by several orders of magnitude due to the coupling of the surface plasmons with the radiation field. In addition, due to a manifold of decay channels for the excitation energy in the metal, fluorescence is efficiently quenched. Serious drawbacks, however, include adsorption-induced and photo-induced degradation of the immobilized proteins. In fact, the first results by CARS and SERRS were not very promising [61,62,63], such that both approaches were not employed anymore in further studies.

In contrast, femtosecond (fs)-stimulated Raman scattering (FSRS), which, like CARS, requires a highly complex experimental setup, is a technique with strong future potential [64,65,66]. In FSRS, a picosecond (ps) probe pulse that controls the spectral resolution is combined with a spectrally broad fs pump pulse governing the time resolution to generate a stimulated Raman spectrum. When studying a protein-bound chromophore, the probe pulse is tuned in resonance with its electronic transition. Then, FSRS selectively probes the chromophore, and due to the coherent character of the stimulated Raman scattering, fluorescence interference can easily be avoided. As the most intriguing advantage, however, the transform limit can be overcome. Thus, it is possible to probe ultrafast processes upon the temporal correlation of the FSRS pulses with an additional fs photolysis pulse that initiates the reaction of interest [64].

The RR techniques described so far were designed to establish resonance conditions with the electronic transition of the tetrapyrrole chromophore. Qualitatively different information can be obtained when the excitation line is shifted to the UV spectral region in resonance with the electronic transitions of the aromatic amino acid residues of the protein [67]. UV-RR spectroscopy probes the intramolecular interactions of Trp and Tyr, but its application of phytochrome suffers from the large number of these amino acids, which makes the interpretation of possible changes very difficult [68,69].

In contrast to IR spectroscopy, which is restricted to the (frozen) solutions of phytochromes, RR spectroscopy can also be applied to phytochrome crystals. Such experiments were carried out with low-energy NIR excitation (1064 nm) to avoid photodestruction [70,71,72]. Thus, it was possible to obtain vibrational spectra from the state of the phytochrome sample for which a crystallographic analysis yielded a three-dimensional (3D) structural model. This is particularly advantageous when the Raman spectra of the chromophore are evaluated by quantum-mechanical/molecular mechanics (QM/MM) calculations that are based on the crystal structure (vide infra) [71].

### 2.3. Spectra Interpretation

In addition to experimental obstacles, the interpretation of the vibrational spectra represents an enormous challenge. IR difference spectra are typically dominated by signals of the amide I and II modes of the protein and the localized C=O stretching modes of the tetrapyrrole substituents. In some cases, the distinction between protein and chromophore bands was not unambiguous a priori but could be clarified by isotopic labeling [73,74,75]. ^13^C-labeling of the apoprotein of the bathy phytochrome Agp2 from *Agrobacterium fabrum* and its reconstitution with unlabeled BV chromophore allowed for the identification of unusual protonated propionic C=O stretching by ruling out the alternative assignment to a carboxyl amino acid side chain [75]. Agp2—like other bathy phytochromes—also allowed for the sequential H/D exchange of the pyrrole N-H groups [75]. Thus, the C=O stretching modes of rings *A* and *D* could be distinguished due to coupling with the N-H co-ordinates of the same rings [75], thereby confirming earlier assignments based on the selective ^18^O-labelling of the ring *A* C=O group [73].

The greatest problem, however, was initially the vibrational assignment of the RR spectra. Until the first crystal structure determination of a phytochrome [14], information about the structure of chromophore was rather vague. Extraction experiments suggested a *ZZZ* and *ZZE* configuration for chromophore in Pr and Pfr, respectively [76]. No direct evidence for the conformation of a methine bridge was available, such that the starting point of a normal mode of analysis, i.e., the molecular geometry, was poorly defined. Thus, one could not build upon the successful empirical normal mode analyses of cyclic tetrapyrroles, i.e., porphyrins [77,78,79].

Instead, emphasis was laid on a spectral comparison using model compounds (e.g., biliverdin dimethyl ester and related tetrapyrrole derivatives) or tetrapyrrole-binding proteins of a known structure and isotopically labeled tetrapyrroles (PCB and derivatives) assembled with apo-phytochromes [51,70,77,78,79,80,81,82,83,84,85,86,87,88,89,90,91,92,93]. However, due to the high conformational flexibility of linear methine-bridged tetrapyrroles, the model compounds could hardly mimic the specific structures of the chromophores in phytochromes stabilized by interactions with surrounding amino acids. Thus, the main benefit of studying the model compound was to use it for testing and training molecules for developing theoretical approaches to calculate vibrational spectra, from semi-empirical to quantum chemical methods [49,51,81,83,84,85,86,92,94,95,96,97,98,99].

Among them, density functional theory (DFT) calculations were the best compromise between computational costs and accuracy. When applying the appropriate scaling procedures, the frequency error could be reduced to less than ±15 cm^−1^ [19,96]. Nevertheless, the inevitable drawbacks were the specific van der Waals and electrostatic interactions with the protein, which affect the structural and electronic properties of the chromophore in a way that could not be mimicked by calculation of the molecule in vacuo. This problem was eventually solved with the development of QM/MM techniques, specifically when combined with molecular dynamics (MDs) simulations [100,101,102]. Here, the segment of interest, i.e., the chromophore and possibly nearby amino acids, are treated quantum mechanically, whereas, for the remainder of the protein, an empirical force field is used. QM/MM requires a reliable 3D structure as the starting point and was successfully applied to tetrapyrrole-binding proteins and eventually to phytochromes first by the Mroginski group [71,103,104,105].

A good reproduction of the RR spectra by the QM/MM calculations can serve as a criterion for the quality of the 3D structure model, not only as far as the chromophore itself is concerned but also with respect to the protonation pattern of the amino acids adjacent to the chromophore [106], which is not accessible by protein crystallography. The spectroscopic-theoretical characterization of a reference state, typically one of the parent states, may then constitute the starting point for the analyses of those states for which, due to the lack of experimentally determined structures, only tentative 3D models are suggested. Here, the comparison between the calculated and experimental RR spectra then guides the refinement of the model. In addition, with an increasing set of data, for instance, derived from snapshots of an MD simulation, structure–spectra relationships can be obtained that allow for estimating the geometrical parameters of tetrapyrrole, even without extensive calculations [105,107].

## 3. Photo-Induced Reaction Mechanism and Chromophore Structural Changes

The reaction sequence of the photo-conversions of phytochromes was first determined on the basis of transient or low-temperature UV-vis absorption spectroscopy [108,109]. The various species are thus distinguished according to differences in the electronic transitions of the chromophore. The resultant scheme seems to be valid for all prokaryotic and eukaryotic phytochromes (Figure 1E).

### 3.1. Parent States

Due to the difficulties described in Section 2.3, the determination of the chromophore structure in the parent states based on the RR spectra yielded only partially correct results. Fodor et al. suggest a *Z* configuration and syn conformation (*Zs*) for the *C-D* methine bridge in Pr, which was assumed to be converted to *E* and *anti* (*Ea*) in Pfr [49]. By extending the QCFF/π calculations to the Lumi-R spectrum, these authors concluded that *C-D* undergoes a *Z* → *E* photoisomerization in Pr, followed by thermal *syn* → *anti* isomerization in the subsequent relaxation to Pfr [54]. Based on DFT calculations, Hildebrandt and Mroginski suggested a *ZZZasa* configuration for Pr instead and a *ZZEasa* and *ZZEssa* configuration for Lumi-R and Pfr, respectively [51,97,99]. The first crystal structures then demonstrated that the chromophore of Pfr was correctly predicted (*ZZEssa*) [15,17], in contrast to Pr for which a *ZZZssa* configuration was found [14], differing by the conformation of a methine bridge single bond proposed in the spectroscopic studies [49,51].

Representative spectra of the parent states of prototypical and bathy phytochromes are shown in Figure 2.

They include several marker bands that are characteristic of specific structural parameters. Of particular interest is the protonation state. In this respect, there was a general agreement already in early RR spectroscopic studies that Pr and Pfr include a chromophore with all pyrrole nitrogens carrying a proton (cationic chromophore) [48,49,51]. The cationic protonation state is reflected by a band between 1580 and 1545 cm^−1^ that originates from a largely pure N-H in-plane bending (NH ip) mode of rings *B* and *C* [51]. Its frequency seems to increase with increasing hydrogen bond strength [71]. Further marker bands refer to chromophore torsion. Mathies et al. pointed out that the bands in the region between 780 and 840 cm^−1^, which are mainly due to the hydrogen out-of-plane (HOOP) modes of the methine bridges, gain RR intensity when there are torsions around the respective double bonds. Correspondingly, the Pfr chromophore with an extraordinarily intense HOOP mode at ca. 820 cm^−1^ was concluded to exhibit a highly distorted *C-D* methine bridge [49].

In addition, a later combined experimental-theoretical study of Pfr demonstrated that the frequency of the *C*-*D* HOOP mode is inversely correlated with the torsional angle of the methine double bond, whereas the C=C stretching frequency of the *C-D* methine bridge (*CD*) increases with the torsional angle of the *C-D* single bond [105]. Similar correlations were derived from the analysis of the Pr spectra [107].

The parent states of a large number of eukaryotic and prokaryotic phytochromes were systematically studied by RR spectroscopy. Notable differences were only identified when comparing prototypical and bathy phytochromes. In Pfr of prototypical phytochromes, the chromophore displays a temperature-dependent conformational equilibrium with two conformers differing with respect to the structures of the *C-D* and the *A-B* methine bridges. This is reflected by the two band components of the *C-D* HOOP and the *A-B* stretching modes (Figure 2) [88,105]. In contrast, such structural heterogeneity does not exist in the Pfr state of Agp2, which appears to be a common property of all bathy phytochromes and may reflect the high thermal stability of this state [75,105,111]. A second unique feature, only observed in the Pfr state of bathy but not of prototypical phytochromes, is the sequential H/D exchange of the pyrrole N-H groups, which occurs instantaneously for rings *A*, *B*, and *C*, but requires many hours for ring *D*, presumably due to the tight salt bridge with the side chain of the highly conserved Asp196 [72]. Accordingly, an immediate H/D exchange also at ring *D* only takes place after the photoisomerization of the chromophore, which destroys the salt bridge [75,111]. The third characteristic property of bathy phytochromes is the propionic side chain of ring *C,* which, in Pfr, remains protonated even up to a pH of 11. Interestingly, the C=O stretching of the protonated side chain is even detectable in the RR spectrum. This suggests that the underlying C=O bond length varies slightly in the electronically excited state of the chromophore [44]. These three structural properties that are readily identified by RR spectroscopy do not hold for the bathy-like phytochrome XccBphP from *Xanthomonas campestris pv. Campestris*, in which the Pfr state prevails at ambient temperature [112]. Here, only the sequential H/D exchange is observed in the Pfr state, whereas the chromophore exhibits a heterogeneous structure, and its propionic side chain is deprotonated [110].

The Pr state of bathy phytochromes has been characterized by RR spectroscopy only in the case of Agp2. Here, the chromophore was shown to form a keto-enol equilibrium which, upon the protonation of the nearby His278 in the last step of the Pfr → Pr photo-conversion, generates a “reactive” enol tautomer for the thermal back isomerization [75]. The RR spectrum of the enol tautomer can easily be confused with a deprotonated tetrapyrrole [113]. The unambiguous identification requires additional IR spectroscopic analysis to confirm the disappearance of the carbonyl function of ring *D* [75].

In most prototypical phytochromes, Pr is not a structurally homogeneous state, in analogy to Pfr. The analysis of the Pr state of Cph1Δ2 by RR spectroscopy indicated that the red absorption band and its shoulder are due to the vibronic transition of a single species [114]. However, the ground state heterogeneity of Pr (also in the case of Cph1Δ2) is due to more subtle structural differences that do not affect electronic transitions to a detectable extent. Such differences include, as is the case in Pfr, the *C-D* and, specifically, the *A-B* methine bridge, as reflected by the pairwise appearance of the same mode (e.g., *A-B* stretching, Figure 2) [106,110,115]. In addition, a recent 2D IR study on *Dr*BphP (*Deinococcus radiodurans*) revealed heterogeneity in the hydrogen bonding environment of the ring C=O group [116].

In the Pr state of Cph1Δ2, heterogeneity is associated with the different protonation states of the nearby His residues (His260, His290) [117]. The transition between these states, which exhibit different RR signatures, occurs between pH 7 and 8. Thus, in most studies, two conformers are present. Interestingly, the Pr crystals of Cph1Δ2, used for X-ray structure analysis [18], were generated from a protein solution buffered below pH 6 and, thus, include only one conformer, as demonstrated by the RR spectroscopic comparison of Cph1Δ2 crystals and solution [71]. Nevertheless, phytochromes may also display a structural heterogeneity in the crystalline state. A RR spectroscopy study on *Dr*BphP and Agp1 revealed that slightly different conformers of the chromophore might coexist in the crystals of the respective Pr states [70].

In terms of the chromophore structure, the parent states are largely robust towards amino acid substitutions in the chromophore binding pocket [53,75,105,115,118,119,120,121,122,123]. In general, the frequency shifts, when compared to the respective wild-type (WT) spectra, are less than 3 cm^−1^. This is also true for the variants in which the substitution impairs the complete photo-conversion between the parent states.

The consequences of the composition of the domain for RR spectra cannot be generalized except for the comparison between the photosensor module and the full-length protein. In these cases, no spectral differences have been detected for plant and bacterial phytochromes [51,75,124]. The truncation of the photosensor module (PAS-GAF-PHY—PGP) to the PAS-GAF (PG) construct in prototypical bacteriophytochromes, which usually abolishes photo-conversion to Pfr, have similarly small effects on the RR spectra of Pr as single-point substitutions, typically restricted to torsional, deformation, and HOOP modes (vide supra) [70,115,119]. In plant phytochromes, however, the spectral differences between PGP and PG are larger and include the vibrational modes in the entire spectral range. This is also true for the Pfr or Pfr-like states that are formed via the irradiation of Pr [125]. Somewhat fewer spectral changes in Pr (but a similarly perturbed photostate) were observed for the tryptic 39-kDa fragment of phyA, which lacks the PHY domain and the N-terminal extension (NTE) [126]. Solely cutting the NTE of the photosensor module of phyB (PGP) mainly perturbs the conformational equilibrium in Pfr, as reflected by the altered intensity distribution between the conjugate HOOP modes [125,127].

Vibrational spectroscopy has also been extended to plant, and cyanobacterial phytochrome adducts with non-native tetrapyrrole chromophores [12,83,87,88,105,107,128]. It was shown that PCB and PΦB could readily be exchanged without significant perturbations to the chromophore structures [3]. In bacterial phytochromes, the chromophore binding site is located in the PAS domain, where only BV can be bound [129]. Upon appropriate amino acid substitutions, however, the chromophore binding site can be shifted to the GAF domain, as is the case in plant phytochromes, and now only PCB or PΦB is covalently attached [130]. The respective RR spectra of the parent states are very similar to the corresponding adducts of plant phytochromes, indicating the interactions in the new chromophore binding pocket do not impose strongly modified electrostatic or steric constraints on tetrapyrrole conformation [88]. Conversely, covalent attachment does not appear to be critical to stabilizing a specific chromophore structure, at least in the case of BV at the natural binding site of bacterial phytochromes in the PAS domain. When the binding Cys20 is exchanged against an Ala, the resultant RR spectra of Pfr only show a slight shift in the equilibrium of the conformational sub-states, as reflected by the altered intensity distribution of the *A-B* stretching modes [105].

Whereas PCB and PΦB are expected to interact in a similar way with the surrounding amino acids in the chromophore binding pocket, the chromophore-protein interactions are qualitatively different when the propionic side chains are esterified [131]. The incorporation of BV derivatives carrying a free and an esterified (methyl) propionic side chain have a different effect on the parent state structures in prototypical Agp1 and bathy Agp2. In Agp1, the RR spectra display a similar vibrational band pattern as the wild-type (WT) protein with minor differences in frequencies and relative intensities. Only a few bands due to tetrapyrrole torsion and N-H in-plane modes show frequency shifts of nearly 10 cm^−1^. In Agp2, this observation also holds for Pfr, whereas, for Pr, the main effect is the complete shift of the tautomeric equilibrium towards the enolic form.

Altogether, the large body of experimental data indicates that the chromophore structures in the parent states of eukaryotic and prokaryotic phytochromes are very similar, as judged by RR spectra. When considering the intrinsic spectral differences due to the different types of tetrapyrroles (PΦB, PCB, or BV), the frequency differences between the same states rarely exceed 15 cm^−1^ and 15% in relative intensities, regardless of the origin of the phytochrome, single or multiple amino acid substitutions, or domain compositions. As an instructive example, we refer to the highly homologous bacterial phytochromes Agp1 and *Dr*BphP [105,132]. The differences in the HOOP and *C-D* stretching mode frequencies in the RR spectra of the Pfr state are 15 and 9 cm^−1^, respectively. In contrast, the fungal phytochrome FphAN753 from *Aspergillus nidulans* displays an RR spectrum that is nearly identical to that of Agp1 with only subtle frequency shifts of 1–2 cm^−1^ for the HOOP and *C-D* stretching modes.

In this context, it is instructive to illustrate the sensitivity of the RR spectra towards structural changes. For instance, a frequency shift of the *C-D* stretching mode of 10 cm^−1^, which in RR spectroscopy is considered to be a large spectral change, corresponds to an increase or decrease of the *C-D* methine bridge tilt angle by only ca. 6°. Such a geometrical change in the chromophore is typically beyond the detection limit, even for well-resolved X-ray structures [107]. However, it should be noted that those structure-frequency correlations discussed in this section only exist for a small number of modes that are, in general, highly localized in a specific internal co-ordinate.

### 3.2. First Events of the Photo-Conversion

The photo-excitation of Pr or Pfr leads to double bond isomerization at the *C-D* methine bridge [133]. Early transient absorption and fluorescence studies of Pr indicated that the first ground state intermediate, Lumi-R, is formed between 30 and 60 ps [109,134,135]. This intermediate of plant phytochrome phyA was cryogenically trapped and characterized by SERDS spectroscopy, demonstrating that thermal activation is required for its formation [54]. The results were later confirmed by cryogenic RR spectroscopy [51], thereby correcting an earlier RR spectroscopic analysis by the same group [136]. RR spectroscopy was further extended to the Lumi-R photoproducts in other phytochromes, revealing remarkable differences between the various species [137].

The most detailed vibrational spectroscopic studies have been carried out with Cph1Δ2 using FSRS and ultrafast transient IR absorption spectroscopy. These techniques allow for monitoring the early events after photo-excitation up to the formation of the Lumi states. The Mathies group employed FSRS to focus on the photo-induced processes of the Pr states [65,66]. After 100 fs, the spectrum shows the signature of the electronically excited state of Pr (Franck Condon state) (Figure 3) with the HOOP, NH ip, and the methine bridge C=C stretching modes downshifted with respect to the ground state of Pr.

This state decays after 200 fs, but the development of the vibrational signature is partially obscured by the signals of a nonlinear effect RINE (Raman induced by nonlinear emission). At ca. 600 fs, an excited state intermediate I* is formed, in which the HOOP mode is still very intense but significantly upshifted compared to the Franck-Condon state. This intermediate decays to the electronic ground state, either returning to Pr (85%) or to Lumi-R (15%) within 40 ps. The spectrum of the latter is similar, albeit not identical, to the first intermediate that could be trapped in cryogenic RR spectroscopy (200 K) [137].

Experiments by the Heyne group provided complementary information by transient IR spectroscopy, monitoring the C=O stretching modes, including their polarization-resolved spectral changes [42]. The rotation of *C-D* methine takes place in the electronically excited Pr state, which then decays to Lumi-R with a time constant of ca. 30 ps, thus being similar to that derived from FSRS [65]. An excited state intermediate I* was not detected [42].

Additional evidence for this intermediate, however, was obtained from multi-pulse transient IR spectroscopic experiments in which, in addition to the visible pump and IR probe pulses, an extra visible dump pulse was also applied to recover the excited states [39,138]. In this study, the IR probe pulses covered a wide spectral range from the C=O to the C=C stretching region. The data were then treated by a global analysis on the basis of different reaction models. The most plausible interpretation was an extension of the scheme derived from the FSRS experiments [65], as the unproductive excited state decay via I* leads first to the ground state of this intermediate, which then relaxes to Pr [138].

Further time-resolved IR spectroscopic measurements were carried out with bacterial phytochromes Agp1, *Rp*BphP2 and *Rp*BphP3 from *Rhodopseudomonas palustris*, *Sa*BphP1 from *Stigmatella aurantiaca*, and *Dr*BphP, including engineered variants [40,41,139,140]. Schuhmann et al. analyzed the photo-induced temporal evolution of the IR absorption of the C=O modes in the Pr state of Agp1 from 300 fs to 100 ps [40]. The authors proposed a scheme similar to that described above for Cph1Δ2 (vide supra). A related approach, but also including polarization-resolved experiments, was applied to Agp1, carrying the natural BV chromophore and a locked BV derivative that could not undergo a photoisomerization [139]. The comparative analysis of the photo-active and photo-inactive adducts helped with sorting out the excited state processes in Pr and the photochemical reaction to Lumi-R. In a study on the photoprocesses of Pr of two related bacteriophytochromes, it was found that, in *Rp*BphP2, the excited state of BV decayed within ca. 60 ps, whereas for *Rp*BphP3, a much slower decay of ca. 360 ps was observed [41]. This difference was attributed to the stronger hydrogen bond of ring *D* in the Pr ground state of *Rp*BphP3, raising the energy barrier for the rotation of the *C-D* double bond. A similar conclusion, i.e., the control of the photochemical conversion efficiency via different hydrogen bonding interaction strengths, was drawn from a comparative study of *Sa*BphP1 and *Dr*BphP variants [140]. Kübel et al. analyzed the changes in the C=O stretching modes during the Pr → Pfr transformation in *Dr*BphP in terms of two Lumi-R intermediates [43]. In their time-resolved IR spectroscopic study, the authors concluded that Tyr263 plays a critical role in controlling the transition from the early to the late Lumi-R.

Ultrafast vibrational spectroscopy was employed to study the photo reactions of the Pfr of Cph1Δ2, Agp1, and Agp2 [44,53,141,142,143,144]. In general, it was found that the formation of the first ground state product Lumi-F occurred much faster than the corresponding reaction of Pr (vide supra), with 1.6 ps in Agp1 and 0.7 ps in Cph1Δ2. In the latter case, both ultrafast IR and FSRS were combined to achieve a comprehensive description of the reaction dynamics up to the formation of Lumi-F. Agp2 is a special case since, here, transient proton transfer upon electronic excitation was detected [44]. While this step, which is ascribed to the deprotonation of ring *C* or *D* and the formation of a nearby protonated water cluster, is reversed with the decay of the excited state, a reorganization of the hydrogen bonding network in the vicinity of the isomerization site develops upon proceeding further along the reaction co-ordinate.

As the main advantage over other optical techniques, ultrafast FSRS and transient IR absorption spectroscopy provide information about the reaction and structural dynamics after photo-excitation. However, the extraction of this information relies upon the sound assignment of the vibrational modes, which—as discussed in Section 2.3.—is difficult, even for the stable states, and is certainly a major challenge for the state of the chromophore that is not even in a (local) energetic minimum along the reaction co-ordinate. A few signals in both the FSRS and the transient IR spectra, however, correspond to highly localized modes, such as the Raman band at ca. 800 cm^−1^ and the IR bands between 1690 and 1740 cm^−1^, which are attributable to the HOOP mode of the *C-D* methine bridges and the C=O stretching modes of rings *A* and *D*, respectively. In combination with the polarization changes of the C=O stretching modes, the series of time-resolved Raman and IR spectra can then be plausibly interpreted in terms of the rotation of the *C-D* methine bridge and ring *D*, accompanied by the exchange of the hydrogen bonding partner of the ring *D* C=O substituent [41,65,138,142,143].

Spectral changes in the C=C stretching region are more difficult to interpret since the normal mode composition is likely to be altered upon photo-excitation when compared to the stable reference state. Thus, the wealth of difference signals in this region and their temporal evolution, for instance, as is shown in the multipulse-IR spectra of van Wilderen et al. [138], still require a reliable approach for their translation into structural data.

Finally, it is interesting to compare the ultrafast time-resolved data with those obtained by cryogenic IR and RR spectroscopy [29,31,51,54,75,122,128,137,145]. A detailed RR spectroscopic study on various phytochromes revealed two Lumi-R states, Lumi-R1 and Lumi-R2, that are presumably formed sequentially. This conclusion is in line with the time-resolved IR spectroscopic experiments by Kübel et al. [43]. The two Lumi-R states appear to be a common mechanistic property of all phytochromes but could not be trapped for each species in a pure form [137]. In general, when compared to Pr, the frequency of the *C-D* HOOP mode is upshifted in Lumi-R1, but this shift is reversed in Lumi-R2. The *C-D* stretching mode shows the opposite tendency since it displays a distinct downshift in Lumi-R1, which is reversed, albeit only partially in Lumi-R2. *A-B* stretching remains largely unchanged. These spectral changes reflect the structural relaxation of the *C-D* methine bridge, whereas the remainder of the chromophore is only slightly affected. According to these general tendencies, one may attribute the state measured by the FSRS of Cph1Δ2 at 40 ps to Lumi-R1, which could not be enriched at the temperatures typically used for trapping Lumi-R1 in other phytochromes (i.e., ca. 100–130 K). In Cph1Δ2, the first photoproduct that was cryotrapped required higher temperatures (200 K) and was identified as the Lumi-R2 state. The failure to trap a Lumi-R1 species at lower temperatures in Cph1Δ2, which was also reported in the IR study by Förstendorf et al. [29], points to a low energy barrier for the transition from Lumi-R1 to Lumi-R2 [137]. Conversely, in phyA, Lumi-R1 is readily trapped at ca. 130 K [137], and the corresponding IR spectra show a distinct frequency upshift of the ring *D* C=O stretching [31,128], which is in line with the results of the time-resolved IR spectroscopy of Cph1Δ2 at 50 ps [42]. Altogether, there seems to be a good correspondence between the time-resolved IR and FSRS spectra (measured up to delays of tenths of ps) and the cryogenic IR and RR spectra on the basis of early and late Lumi-R1 and Lumi-R2 photo-products.

An early and a late Lumi-F could not be directly distinguished in the photo-conversion of Pfr. However, on the basis of selective H/D exchange effects on the RR spectra, it was shown that only at elevated temperatures but prior to the thermal decay to Meta-F, the Lumi-F of bathy phytochromes undergoes a photochemical backreaction to a Pfr-like state (Pfr‘) that eventually decays to Pfr [75,137]. This branching was not observed at lower temperatures, which suggests either a thermally activated photochemical reaction sequence back to Pfr or two Lumi-F states with different photochemical reactivities.

### 3.3. Late Events of the Photo-Conversion

Whereas the conformational relaxations in the Lumi states are restricted to chromophores and the directly interacting amino acids, the subsequent thermal reaction steps include protein on different levels of the structural hierarchy. Here, there are also notable differences in the reaction mechanism between Pr → Pfr and Pfr → Pr photo-conversions, as well as between prototypical and bathy phytochromes. The sequence of the cryo-trapped intermediates following Lumi-R are Meta-Ra, Meta-Rc, and Pfr for the Pr → Pfr photo-conversion of both prototypical and bathy phytochromes. The RR spectroscopic characterization of Meta-Ra in all phytochromes studied so far revealed a protonated chromophore and low HOOP activities corresponding to sterically relaxed methine bridges [51,110,122,137]. The frequency of the *C-D* stretching mode is upshifted compared to the Lumi-R states and is, again, close to or even higher than that of Pr. The subsequent intermediate is Meta-Rc, in which either ring *B* or ring *C* is deprotonated. Chromophore deprotonation has originally been proposed on the basis of RR spectra excited in resonance with the second electronic transition [56,57]. This conclusion was initially questioned by the Hildebrandt group [51] but later confirmed for plant as well as bacterial phytochromes [52,70,110,118,119,137]. The release of the proton from the chromophore is linked to the transfer of a proton to the solution phase, as shown for Agp1 and Cph1Δ2 [52,117,118]. This proton translocation is reversible, and the chromophore is reprotonated with the formation of Pfr. However, the postulated intermediate with a reprotonated chromophore preceding Pfr has been trapped and RR spectroscopically characterized only in one case of a plant phytochrome variant [125].

The protonation state changes of the chromophore are more difficult to identify in the IR difference spectra since the characteristic bands of the NH ip bending modes are expected in (spectrally) rather cogent regions, including difference signals from the protein [29,73,74,128,145]. Although distinguishing between the protein and chromophore bands was facilitated by using isotopically labeled chromophores [74,128], an unambiguous assignment of the protonation marker bands in the IR difference spectra was not possible. However, in all studies, it was consistently observed that the main protein structural changes occur in the last step of the photo-conversion [29,31,128,145]. After the photo-induced changes of the protein’s secondary structure were identified via X-ray crystallography [146], the signals in the amide I band region of the IR differences spectra could readily be assigned to the tongue segment of the PHY domain [53,75,131,147].

On the basis of the vibrational assignments, time-resolved IR spectroscopy provides an excellent tool to elucidate the structural and reaction dynamics of phytochromes [36,43,45,140,148]. Using the step-scan technique, Kottke and colleagues probed the spectral changes of the Pr → Pfr photo-conversion in *Dr*BphP PGP monomers [148]. The representative spectra in Figure 4 show the characteristic features of Lumi-R, Meta-Ra, and Pfr at 77 μs, 1.46 ms, >5 ms, respectively. The temporal evolution of the corresponding species-associated difference spectra agrees very well with the data obtained from a flash-photolysis experiment carried out in the visible range.

### 3.4. Thermal Back Reactions

The parent states Pr and Pfr are linked via both photochemical and thermal reaction pathways. Little is known about the thermal route since the first steps must be associated with high energy barriers, i.e., the double bond isomerization and the secondary structure transition of the tongue. Thus, the subsequent reaction steps are likely to be much faster, which can explain why the identification of intermediates failed. A key requirement for the thermal isomerization, as postulated by Lagarias and Rapoport [149], is the transient formation of an enolic structure of the chromophore, which may reverse the order of the double and single bonds at the *C-D* methine bridge. In fact, such a structure has only been identified IR and RR spectroscopically for the Pr state of the bathy phytochrome Agp2 [75]. In that protein, the enolic structure requires specific interactions with the protonated His278, as originally proposed by Velasquez et al. [75] and further refined by Lopez et al. [110]. Up to now, no enolic tautomer has been detected in any other phytochrome state. This may be due to the low RR scattering cross-section upon excitation in resonance with the first electronic transition and the low steady-state concentration of a reactive tautomer.

Recently, time-resolved RR and IR spectroscopy revealed that the Meta-F and Meta-Rc states of Agp2 and Agp1 are also capable of undergoing a thermal back reaction [150,151]. This branching mechanism appears to be a general property of all phytochromes and represents a shortcut for photo-conversion, corresponding to the recovery of the parent state without inducing the functionally important structure transition of the tongue. In Agp2, the ratio between the unproductive (Meta-F → Pfr) and productive (Meta-F → Pr → Pfr) branch is ca. 1:1.

## 4. Coupling of Chromophore and Protein Structural Changes

Protein structural changes follow photo-induced chromophore structural changes sequentially on different levels of hierarchy. First, during the lifetime of the Lumi states (vide supra), there are low-amplitude re-reorientations of the amino acid side chains and water molecules at the isomerization site. These events are followed by the conformational relaxation of the chromophore and the concomitant structural adaptation of the chromophore binding pocket in the Meta states, exemplarily illustrated in a combined crystallographic and spectroscopic study on an Agp2 variant [72]. The Meta states represent crucial intermediates since they are the starting points for a cascade of major structural changes in the protein that transmit the light signal to the output module far away from the chromophore. The first step is the secondary structure transition of the tongue in the PHY domain.

### 4.1. Proton Transfer and Tongue Restructuring

A systematic RR and IR spectroscopic study was carried out on Agp2 to identify the parameters that control the transition of the tongue from α-helix to β-sheet during the reaction step from Meta-F to the final product Pr [53]. It was shown that this process requires the deprotonation of the prop*C* side chain. In WT Agp2, this proton transfer occurs from prop*C* to His278, but this residue can also be replaced by another proton acceptor [75]. The results obtained from various variants indicate a distinct relationship between the amplitude of the amide I signal, reflecting the extent of secondary structure changes, and the degree of the deprotonation of prop*C*, which, in turn, is related to its p*K*_a_ (Figure 5) [53]. Whereas the α-helix to β-sheet conversion in Agp2 and other bathy phytochromes evidently requires a proton translocation inside the chromophore-binding pocket [53,110], the reverse reaction from Pr to Pfr, including β-sheet-to-α-helix conversion, is coupled to the reversible deprotonation of pyrrole ring *B* or *C* and leads to the transport of a proton to the solution phase [52,117,152]. This process starts with the formation of the Meta-Rc intermediate, and reprotonation occurs with Meta-Rc decay. The reversible deprotonation of the pyrrole rings was observed in both prototypical and bathy phytochromes [110]. A proton translocation was not yet identified in the rarely studied Pfr-to-Pr conversion of prototypical phytochromes.

Nevertheless, the findings described above suggest that the coupling of the photo-induced chromophore structural changes with the secondary structure conversion of the tongue is mediated by proton transfer either within the chromophore binding pocket or between the chromophore and the solution. Generally, proton transfer processes correspond to an alteration in the internal charge distribution and, thus, to changes in the electrostatics, which, hence, was suggested to be the driving force for the secondary structure transition [53]. In fact, this hypothesis is consistent with theoretical calculations on model peptides demonstrating that electric field strengths of 10^8^–10^9^ V/m, which may locally exist in proteins [153], can readily cause a transformation from a β-sheet to an α-helix structure [154].

### 4.2. Electric Field Effects

A well-established approach to determining electric fields in protein is based on the vibrational Stark effect (VSE), pioneered in theory and practice by the Boxer group [153]. The VSE refers to the effect of a strong electric field on a vibrational transition, resulting in a shift in the frequency when compared to the absence of the electric field. The VSE is particularly large for localized modes, such as the stretching of C=O and C≡N groups, and can be probed by IR spectroscopy. The nitrile group offers the additional advantage of its stretching mode appearing in a spectral range free from any of the other bands of a protein, such that it is an ideal VSE reporter group to be inserted into proteins [155]. A drawback, however, is the sensitivity of its stretching frequency to hydrogen bonding interactions, which counteracts the VSE and, thus, makes the interpretation of the experimental data rather difficult [153,156,157]. Nitrile groups can be site-specifically incorporated into proteins either by chemical modifications to cysteine side chains [158] or by introducing non-natural amino acids with nitrile function via reprogramming the genetic code [159,160].

Kraskov et al. employed the latter approach in Agp2 to substitute Tyr165 and Phe192 via para-cyano-phenylalanine (pCNF) [123]. Both amino acids are close to the chromophore and not far from (Phe192) or even directly involved in the proton transfer pathway (Tyr165). The RR, IR, and UV-vis absorption spectroscopic characterization of the two variants revealed far-reaching similarities to WT protein, including the deprotonation of prop*C* and the secondary structure conversion of the tongue in the final step of the Pfr → Pr conversion. Thus, it was concluded that the VSE reporter group was not invasive and did not perturb the chromophore and protein structure in F192pCNF and Y165pCNF. As demonstrated by modeling the structures, the two variants represent two limiting cases of a nitrile group in a completely hydrophobic environment (F192pCNF) and strong hydrogen bonding interactions (Y165pCNF), which is reflected by characteristic frequencies below and above ca. 2235 cm^−1^, respectively [156].

IR spectra in the nitrile stretching region were measured for Lumi-F, Meta-F, and Pr and, thus, at different temperatures (Figure 6) [123]. In order to account for the inherent temperature dependence of the nitrile stretching frequency, in each case, the dark spectrum was measured at the same temperature as well, as a reference. For the non-hydrogen-bonded nitrile in F192pCNF, the largest change in frequency (compared to Pfr) was observed in Pr after the proton transfer and restructuring of the tongue. This is also true for Y165pCNF, although, in that case, there is also a quite remarkable frequency shift in Lumi-F.

Assuming the theoretical value for the Stark tuning rate is $\left|\vec{\Delta \mu }\right|=0.61\;{\left(\mathrm{MV}/\mathrm{cm}\right)}^{-1}\cdot {\mathrm{cm}}^{-1}$, which relates the frequency shift with the electric field strength [153], one may estimate the change in the local electric field in Pr with respect to Pfr experienced by the nitrile probe. For F192pCNF, one obtains a value of ca. 6 MV/cm ($6\cdot 10^{8}V/m$), and a similar value (ca. 7 MV/cm) is estimated for Y165pCNF, assuming that the hydrogen bonding interactions of the nitrile group are the same in Pr and Pfr. This is, in fact, the order of magnitude predicted to be capable of inducing the secondary structure transitions of peptide segments. However, it should be considered that these data refer to the local electric field at the reporter groups and do not allow for estimating the relevant electric field changes along the α-helical axis of the tongue [154]. Nevertheless, the reported results are important since they show that such large changes in the electric field may occur in the protein.

In this context, it is interesting to refer to the time-resolved IR spectroscopic study by Kurttila et al., who introduced *para*-azidophenylalanine at different positions in *Dr*BphP [161]. The frequencies of the azide stretching modes cannot be directly related to the electric fields but may be considered as qualitative measures for the polarity of the surroundings [155,162]. Nevertheless, the downshifts in the azide stretching modes, which were found for *Dr*BphP, point to increased polarity [161]. These shifts were observed in the last step of the Pr → Pfr photo-conversion and were also noted for azide groups at amino acid positions are analogous to Phe192 and Tyr165 in Agp2, as well as for a position in the tongue region and thus corroborating the findings and hypothesis by Kraskov et al. [123].

Furthermore, the importance of intramolecular electrostatic fields for protein structural changes was also demonstrated by RR and IR spectroscopic studies of Agp1 and Agp2 harboring a BV monomethylester [131]. In these experiments, it was shown that both Agp1 and Agp2 require a free prop*C* side chain to undergo photo-conversion. In Agp2, deprotonation of prop*C* then takes place as it does in the WT protein (vide supra), but the tongue remains in its α-helical structure despite proton transfer. Evidently, proton transfer alone is not sufficient to trigger secondary structure changes in the tongue; it requires appropriate overall electrostatics, which in this case, was severely disturbed by the esterification of prop*B* [131]. Conversely, the charge neutralization at prop*B* does not impair the β-sheet-to-α-helix conversion in the Pfr of Agp1, and the thermal recovery of Pr is even accelerated. These findings suggest different mechanisms in photo-induced protein structural changes in the Pfr → Pr and Pr → Pfr conversion of bathy and prototypical phytochromes, respectively. Possibly, in the latter case, the secondary structure changes of the tongue are stabilized by dipole-dipole interactions between ring *D* C=O and α-helical tongue conformation, as shown by a recent 2D-IR spectroscopic study on Agp1 [46].

Altogether, the investigation of electrostatic effects on the coupling between chromophore and protein structural changes requires further comprehensive experimental and theoretical investigation. Despite the importance of such effects, it is most likely an oversimplification to consider electric field changes as the only basis of structural communication within the protein [110].

### 4.3. Protein–Protein Interactions

Phytochromes form dimers, and one may intuitively think that upon photo-excitation, both monomers act in parallel, leading to a symmetric activated state. However, it has been recently argued that *Idiomarina* sp. A28L bacteriophytochrome (*Is*PadC), including a diguanylyl cyclase output module, is physiologically active in an asymmetric quaternary structure [163]. In a study combining crystallography, mass spectrometry, and RR spectroscopy, it was, in fact, shown, upon light absorption, that an equilibrium between a Pfr and Pfr-like state is formed [164]. The Pfr-like state combines the features of the Pfr and Meta-R states, as revealed by RR spectroscopy. This state is stabilized by the specific rearrangement of the N-terminal segment. In this way, the PHY-tongue conformation of *Is*PadC is partially uncoupled from the initial changes in the N-terminal segment, and this uncoupling allows for signal transduction to the covalently linked output module.

## 5. Cyanobacteriochromes

Like canonical phytochromes, CBCRs are composed of a photosensor and an enzymatic output module. These phytochrome-like photoreceptors frequently catalyze the production or degradation of cyclic dinucleotides [6]. CBCRs are found in cyanobacteria and carry phycoviolobilin (PVB) or, in most cases, PCB as the chromophore attached to a Cys in the GAF domain of the photosensor module. This module lacks the PHY and PAS domains but may include one or several GAF domains. The chromophores in the parental states of the photoswitch are usually in the *ZZZssa* and *ZZEssa* configurations.

Among the simplest CBCRs are *Sy*A-Cph1 and *Sy*B-Cph1 from *Synechocystis PCC 6803* (PAS-GAF), which undergo classical Pr/Pfr photo-conversion, as shown by UV-vis absorption and RR spectroscopy [165]. Nevertheless, the chromophore geometries are slightly different compared to the related Cph1Δ2 (PAS-GAF-PHY, also denoted *Syn*-Cph1 in that work), as indicated by the shifts in the *C-D* stretching region of the RR spectrum. Additionally, the photosensory module of Cph2 (*Syn*Cph2), including two GAF domains, shows similarly large spectral differences when compared to Cph1Δ2, with frequency differences of up to 7 cm^−1^ in the C=C stretching and HOOP region [166]. When compared to Cph1, these differences may partly be ascribed to a shift in the conformational equilibrium and to structural alterations in the individual conformers.

The most remarkable property of CBCRs, however, is the broad spectral range of the absorption maxima of the two parent states, spanning from near IR to near UV [167,168]. One example is AnPixJ from *Anabaena* sp. PCC 7120 [169]. It is a multi-GAF domain CBCR, with the second GAF domain carrying the PCB chromophore (AnPixJg2). This CBCR switches from a red-absorbing Pr (648 nm) to a green-absorbing Pg (543 nm) [170]. The RR spectroscopic investigation revealed far-reaching similarities regarding the Pr state and those of the canonical phytochromes [170], which is consistent with the crystal structure data [169]. This is also true for the mechanistic pattern of the photo-induced Pr → Pg reaction pathways. The first two intermediates, Lumi-R and Meta-R1, were found to carry a protonated chromophore, whereas the final intermediate, Meta-R2, i.e., the precursor of Pg, has a deprotonated chromophore. A very similar scheme was identified for a photo-induced back reaction, implying that the photo-induced conversions in both directions involve transient chromophore deprotonation, unlike canonical phytochromes [110]. The RR spectra of the Pg state reveal a protonated chromophore [170], which, however, was found to be in conflict with MAS-NMR spectroscopy, indicating a deprotonated PCB [171]. This contradiction could be resolved in a joint study of the Hildebrandt and Matysik groups, who then showed that the in vitro assembly of the protein, which is required for producing the isotopically labeled NMR sample, yielded a deprotonated chromophore in the Pg state in contrast to the in vivo assembly [171], which was used in RR spectroscopic experiments [170] and also in the crystallographic study [169]. Most surprisingly, the chromophore of the in vitro-assembled Pg remained deprotonated over a wide pH range but became protonated when the pH was raised above (!) pH 10 [171]. The most likely explanation is a protein misfolding in the chromophore binding pocket that alters the local electrostatics and, thus, the p*K*_a_ of an amino acid side chain close to the chromophore. Upon raising the pH above 10, the misfolding was repaired, and the chromophore protonated. Regardless of this assembly artifact, the origin of the green absorption was discussed [171]. In comparison to the Pfr state of canonical phytochromes that exhibit the same *ZZEssa* configuration, the RR spectrum of the Pg state of AnPixJg2 displays a distinctly upshifted *C-D* stretching frequency. This was initially attributed to a change in the polarity of the chromophore’s surroundings [170], which is analogous to the well-known inverse relationship between the C=C stretching frequency (in cm^−1^) and absorption maximum (in nm) established for retinal chromophores [172]. However, the high *C-D* stretching frequency may also reflect an increased torsional angle at this methine bridge, which, in turn, reduces the effective length of the conjugated π-electron system [171]. This explanation seems to be more likely in view of the findings of Lagarias and coworkers on related CBCRs [173,174].

The same color tuning mechanism seems to hold for Slr1393, a CBCR including three GAF domains isolated from *Synechocystis* sp. PCC6803 [175]. Slr1393 is a red-green CBCR similar to AnPixJ, and also the RR spectra of the Pr states of both CBCRs are closely related (Figure 7) [176]. More differences were observed for the Pg state, which exhibits a relatively broad band envelope in the C=C stretching region. On the basis of the crystal structure data [175], QM/MM calculations were carried out using 12 snapshots that were taken from different points of an MD production run [176]. The resultant Raman spectra were averaged and provided a good description of the experimental Raman spectra of Pr and Pg (Figure 7: black traces). This is also true for the intermediate state O600, for which the crystal structure analysis revealed a *ZZZssa* chromophore configuration of Pr, albeit with larger torsions at the *A-B* and *C-D* methine bridges [175].

In addition, an β-facial attachment to the protein was suggested. However, a good reproduction of the Raman spectrum could be only achieved with an α-facial attachment (as in Pr) but with essentially the same chromophore geometry as in the experimental structure (Figure 7) [176]. Temperature-dependent RR and UV-vis absorption spectroscopy revealed a conformational equilibrium between O600 and Pr, with the latter prevailing at ambient temperature. Accordingly, the authors described Slr1393 as a light- and temperature-sensitive sensor, although the range with notable temperature-induced color changes (i.e., shift in the Pr/O600 equilibrium) is below 0 °C and is, thus, possibly not physiologically relevant. Color tuning in Slr1393 can readily be understood on the basis of the length of the conjugated π-electron system, which is controlled by the torsion of the methine bridges, specifically the *C-D* methine bridge. This conclusion, which was derived from the analysis of the electronic transitions [175], was found to be consistent with the RR spectra.

Insight into the dynamics of the photo-conversion was obtained by time-resolved IR spectroscopy. Buhrke et al. analyzed the photo-conversions in both directions and could identify two ground state intermediates for the Pr-to-Pg transformation and four species on the reverse photo-conversion route [177].

These authors also studied another red-green CBCR Am1-c0023g2 from the cyanobacterium *Acarychloris marina* by 2D-IR spectroscopy. By using isotopically labeled protein, they succeeded in analyzing the band shapes of the C=O stretching modes of the PCB chromophore, which allows for conclusions about changes in environmental interactions [178].

In contrast to Slr1393 and AnpixJ, the CBCR TePixJ from *Thermosynechococcus elongatus* employs another mechanism to shift the electronic transitions to higher energies. TePixJ represents a green-blue photoswitch with absorption maxima at 534 (Pg) and 430 nm (Pb) [179]. Experimental studies using NMR and vibrational spectroscopies demonstrated a PVB chromophore attached via an additional Cys to the protein, thereby reducing the conjugated π-electron system, although the assignment of the *A-B* methine bridge as the binding site for the second Cys was not correct [179]. Instead, binding occurs at the *B-C* methine bridge in the blue-absorbing state Pb (*ZZZssa*), but this bond is disrupted after photoisomerization in the Pg state (*ZZEssa*) [169,180]. Scrutton´s group studied the photocycles of TePixJ and a related CBCR (Tlr0924) by time-resolved IR and UV-vis absorption spectroscopy from picoseconds to seconds [37,38]. Tlr0924 is a particular challenge for spectroscopists since it contains both a PCB and a PVB chromophore, which run through parallel photocycles, including the reversible binding of a second Cys to the *B-C* methine bridge [37]. The authors employed an elegant approach to separate the photocycles and the contributions to the transient UV-vis and IR spectra (Figure 8). Both PCB and PVB form a blue-absorbing “double-Cys” adduct in the *ZZZssa* configuration. Light absorption causes photoisomerization to a *ZZEssa* configuration and the loss of the Cys attachment to the *B-C* methine bridge. As a consequence, the PCB photocycle leads to a red-absorbing species, and the PVB photocycle leads to a green-absorbing species, which can be selectively pumped back to the originally dark state by red (PCB) and green light (PVB). In this way, it is possible to probe either the PCB or the PVB photocycle.

Another interesting CBCR representative is Oscil6304_2705 from the cyanobacterium *Oscillatoria acuminata* PCC 6304, which is a photoswitch between the blue-absorbing Pb (*ZZZssa*) and the orange-absorbing Po state (*ZZEssa*). The groups of Hirose and Unno employed UV-vis absorption and RR spectroscopy, demonstrating the loss of Cys ligation to the *B-C* methine bridge upon the photo-conversion of Pb [181]. Based on pH-dependent measurements, it was shown that above pH 10, the chromophore of the Po state deprotonates and thermally isomerizes to a green-absorbing species (Pg, *ZZZssa*). It was proposed that Pg is an intermediate on the photo-conversion route from Po to Pb. The same groupS extended the RR studies to RcaE from *Fremyella diplosiphon*, which switches between a red- and green-absorbing state, corresponding to *Z*-to-*E* photoisomerization at the *C-D* methine bridge [182,183]. Supported by QM/MM calculations, the authors concluded that the chromophore in Pg is deprotonated [180]. Furthermore, the RR spectra of the Pr state were found to be consistent with the unusual *syn* conformation of the *C-D* methine bridge, in contrast to the typical *anti* conformation in other CBBRs and canonical phytochromes [181].

## 6. Fluorescing Phytochromes

The vivid development of optical microscopic techniques spurred the search for fluorescing proteins, which could be co-expressed in target cells for in vivo imaging. An important requirement was the emission in the red or NIR region to ensure high penetration depth. Phytochrome chromophores have been identified as promising candidates [184], although the fluorescence quantum yield in their native protein environment is very low. Thus, attempts were made to engineer bacterial BV-containing phytochromes toward increased fluorescence using a combination of selected site-specific and random mutagenesis [185]. In many studies, the PG domain construct of the bacteriophytochrome *Rp*BphP2 from *Rhodopseudomas palustris* served as a starting point. Thus, this protein and its fluorescence-optimized variants were studied by RR spectroscopy in an attempt to elucidate the molecular basis for the increased fluorescence, ultimately as a step towards the rational design of highly efficient fluorophores [115,120,121]. The first fluorescing phytochrome that was analyzed in this way was generated from the D202H variant of *Rp*BphP2(PG) by random mutagenesis, leading to 13 substitutions compared to the wild-type PG construct, including five in the chromophore binding pocket [115]. Supported by MD simulations, the authors identified two factors contributing to enhanced fluorescence: increased rigidity of the chromophore embedment in the protein and a higher degree of ring *D* tilting that reduced the hydrogen bonding interactions with the surrounding amino acids. In subsequent work, the authors analyzed the role of individual amino acid substitutions in a related variant [120]. Two groups of mutations have qualitatively different effects on the chromophore, which exists in a fluorescent (state II) and a non-fluorescent conformer (state I). Three substitutions in the vicinity of the chromophore increased the intrinsic fluorescence quantum yield of state II due to small structural changes in the bound BV, whereas multiple substitutions more remote from the chromophore caused a major shift in the conformational equilibrium towards the fluorescing state.

In another variant derived from *Rp*BphP2(PG), an additional Cys was introduced to provide two thioether bonds to ring *A*, thereby eliminating all the endo- and exo-cyclic double bonds of this ring. In a RR spectroscopic study, Buhrke et al. demonstrated that the particularly high fluorescence quantum yield (ca. 16%) originates from the double attachment of BV. The consequences of each Cys binding on the chromophore structure were found to be additive, with each of them decreasing the flexibility of the chromophore in the protein, as reflected by the effect on the characteristic RR marker bands [121].

## 7. Conclusions

The present review described the contributions of vibrational spectroscopy to the understanding of the structure, dynamics, and function of phytochromes. The impact of the Raman and IR spectroscopic approaches has increased over the past 40 years, not only with technical developments to the methods but also with the progress made in other methodologies, ranging from protein engineering to structural biology and theory. RR and IR difference spectroscopies are selective and, thus, focus on specific structural aspects, such as hydrogen bonding interactions or the conformational details of the chromophore. Accordingly, these techniques are ideal to complement crystallographic methods, which provide an overall 3D picture but generally lack the resolution for the fine structure. Moreover, despite the recent advancement in time-resolved and cryotrap crystallography, the characterization of photo-conversion intermediates mainly relies upon RR and IR spectroscopy. Nevertheless, the full exploitation of the potential of vibrational spectroscopies is only possible if they are applied in combination with other experimental and theoretical approaches.

## Figures and Tables

**Figure 2 biomolecules-13-01007-f002:**
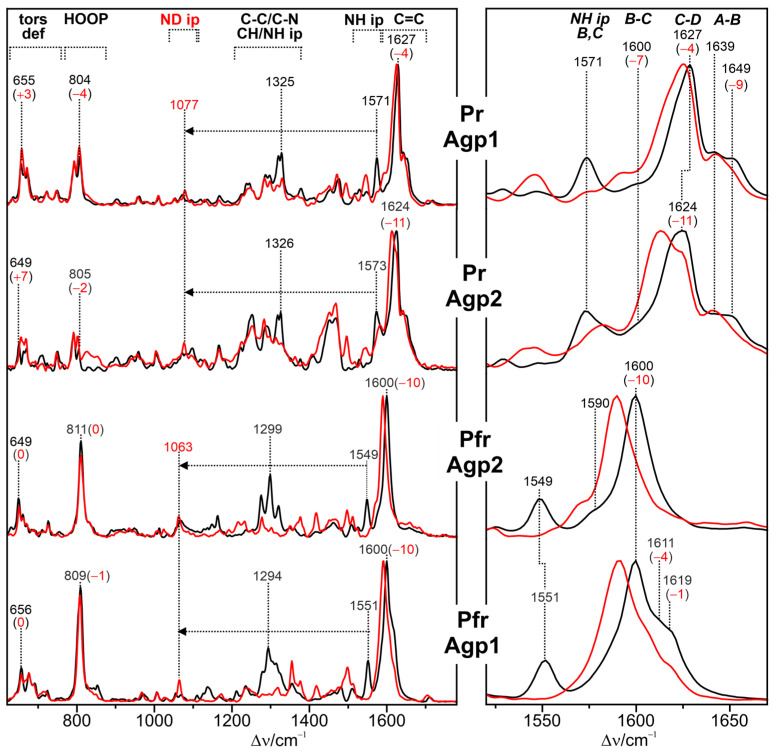
The RR spectra of the parent states Pr and Pfr of the prototypical phytochrome Agp1 and bathy phytochrome Agp2 from *A. fabrum*. The spectra were measured with 1064 nm at 80 K in H*_2_*O (black) and D_2_O buffers (red). Left panel: overview of the spectral region, which is of primary interest for the RR spectroscopic characterization of the chromophore. The main character of the modes in specific regions is described by torsional (tors), deformation (def), in-plane bending (ip), single-bond, and double-bond stretching (C-C/C-N, C=C) co-ordinates. The peak labels refer to the frequencies in H_2_O (black) with the corresponding shifts in D_2_O (red). Only for the largely pure N-H ip modes of rings *B* and *C* are the absolute frequencies given in both H_2_O (NH ip) and D_2_O (ND ip). Right panel: Expanded view of the region of the C=C stretching modes of the methine bridges between rings *A* and *B* (*AB*), *B* and *C* (*BC*), and *C* and *D* (*CD*). Note that, except for Pfr of Agp2, all spectra show two band components originating from *A-B* stretching, thus reflecting the structural heterogeneity of the chromophore. Similar effects are detectable for the HOOP of the *C-D* methine bridge. This region also includes the NH ip modes of rings *B* and *C*. Further details and a comprehensive spectra interpretation are given in references [75,105,106,110].

**Figure 3 biomolecules-13-01007-f003:**
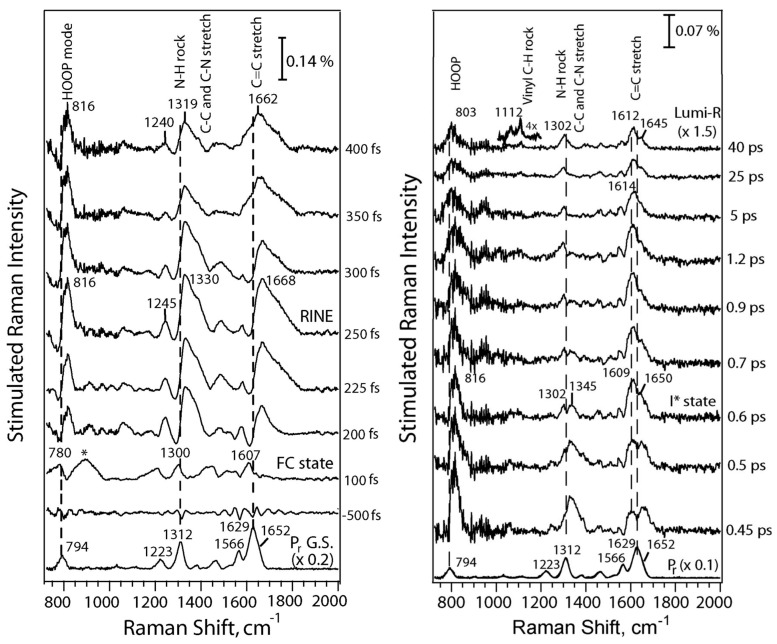
Time-resolved FSRS spectra of Cph1Δ2 were measured by using a 792-nm Raman pump pulse after the photo-excitation of Pr at 635 nm. The left and right panel shows the spectra at the time scales of ≤400 fs and ≥450 fs, respectively. Dispersive peaks from 200 to 350 fs are attributed to hot luminescence features, for which the correspondence to the main ground-state peaks is indicated by the solid vertical lines at 794, 1312, and 1629 cm^−1^. The electronic echo artifact is marked by an asterisk. The Pr ground-state Raman spectrum is scaled as 0.2 (left panel) and 0.1 (right panel). The 40-ps (Lumi-R) spectrum is blown up to 1.5. Reprinted from Dasgupta et al. [65].

**Figure 4 biomolecules-13-01007-f004:**
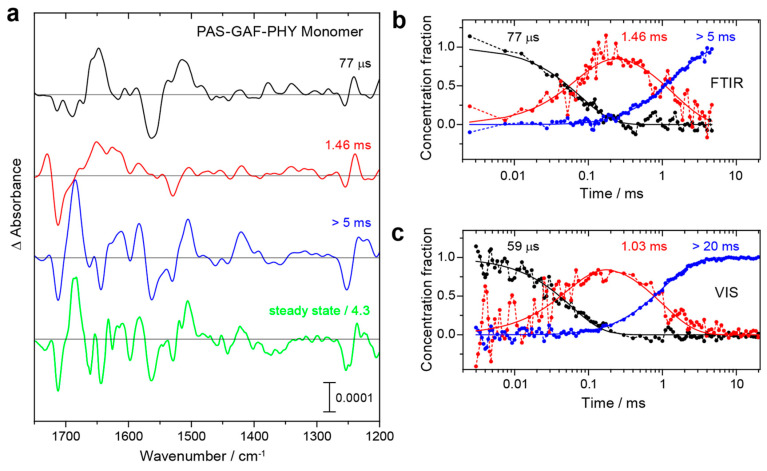
Time-resolved IR spectra of the photo-induced processes of monomeric DrBphP (PGP). (**a**) A global analysis of the step scan IR spectroscopic data of DrBphP yields species-associated difference spectra (SADS) for three components. The steady-state IR difference spectra of Pfr minus Pr are included at the bottom (green). The positive and negative signals are due to the intermediate/product and the initial Pr state, respectively. The spectral resolution of the steady-state spectra is higher (2 cm^−1^) than those from the step scan (8 cm^−1^). (**b**) The concentration fraction of each species was derived as a function of time from the global fit of the IR data. (**c**) A global analysis of the flash photolysis data of dimeric PGP yields the evolution of the three species in the visible spectral range for comparison. Reprinted from Ihalainen et al. [148].

**Figure 5 biomolecules-13-01007-f005:**
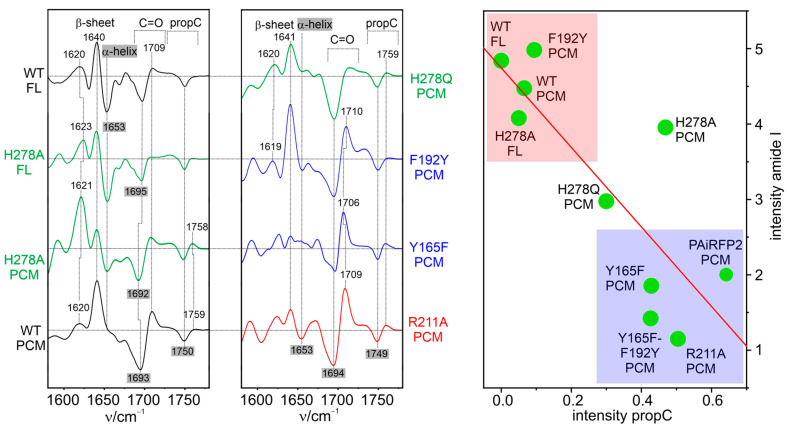
Left panel: IR difference spectra of the final photoproducts (positive signals/black labels) of Pfr (negative signals/black labels on grey areas) of various Agp2 variants, measured at ambient temperature (H_2_O, pH 7.8). The photoproduct spectra were recorded under continuous 785 nm irradiation (>120 s). “FL” and “PCM” refer to the full-length and photosensor core module, respectively. All spectra were normalized with respect to the negative signal at 1749 (1750) cm^−1^. Right panel: plot of the amide I difference signal (β-hairpin + α-helix) against the intensity of the protonated prop*C* of the photo-products determined from the IR difference spectra. The intensities were normalized with respect to that of the protonated prop*C* of Pfr. The straight red line serves to guide the eye but does not implicate a linear relationship. Figure 5 was redrawn from the data shown by Kraskov et al. [53].

**Figure 6 biomolecules-13-01007-f006:**
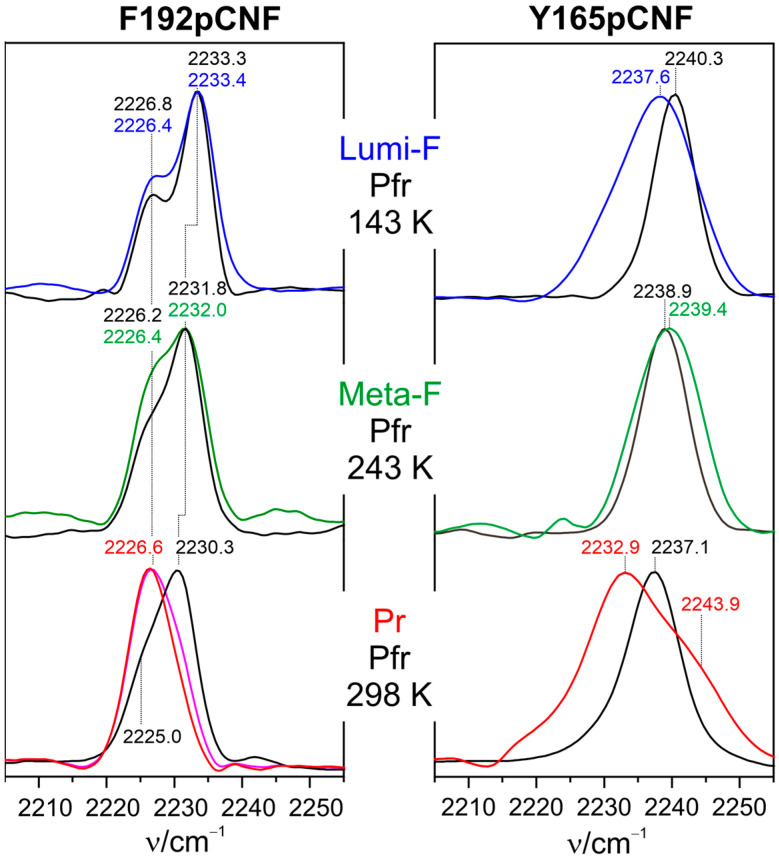
IR spectra of the nitrile stretching modes of F192pCNF (left) and Y165pCNF (right) for the different states (blue, green, and red traces) of the Pfr → Pr photo-conversion, measured at different temperatures. In each case, the spectrum of the Pfr state (black trace) is shown as a reference. To obtain the pure spectrum of Pr of F192pCNF (red trace), a residual contribution (ca. 25%) of Pfr was subtracted from the spectrum measured at ambient temperature (magenta trace). The band positions were determined from fits of Gaussian band shapes to the spectra. Reprinted from Kraskov et al. [123].

**Figure 7 biomolecules-13-01007-f007:**
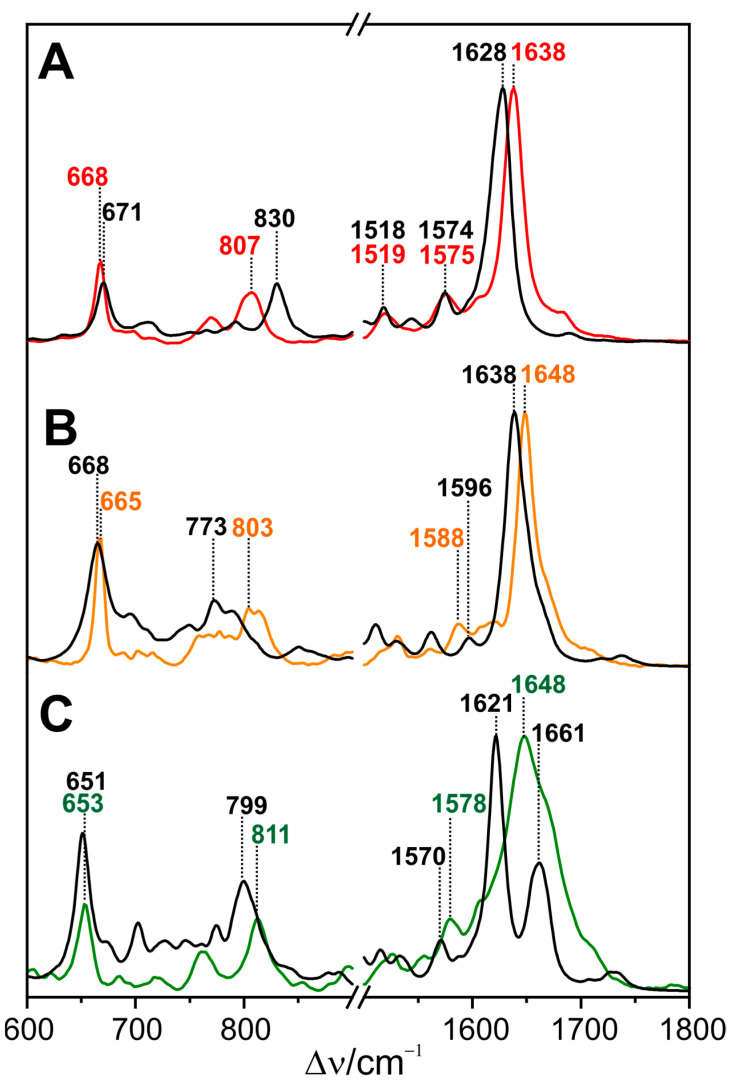
RR spectra obtained with 1064 nm excitation of (**A**) Pr (red line), (**B**) the O600 intermediate (orange line), and (**C**) Pg (green line). The black traces refer to the calculated spectra obtained by the QM/MM calculations. Figure 7 was redrawn from the data shown by Buhrke et al. [176].

**Figure 8 biomolecules-13-01007-f008:**
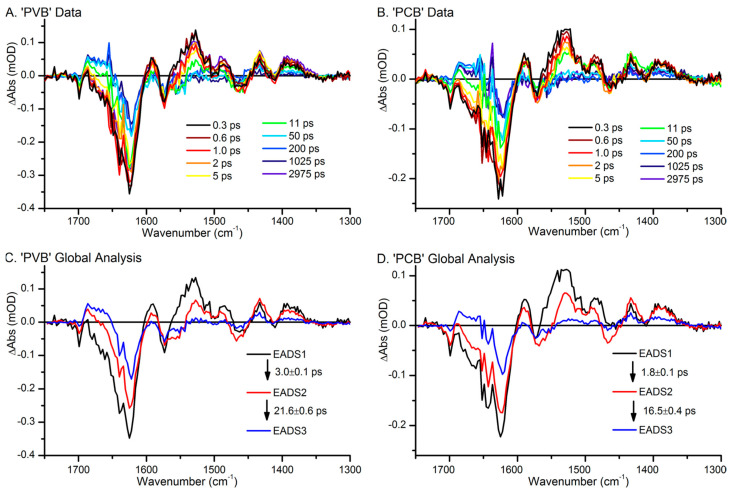
(**A**) transient IR absorption spectra, collected after excitation at 435 nm, at selected time points for “PVB” samples, where the Pb states were constantly regenerated with green light; (**B**) the same for “PCB” samples, where the Pb states were constantly regenerated with red light. The global analyses of the transient absorption data for the “PVB” samples (**C**) and “PCB” samples, (**D**) which yielded three EADS that sequentially interconvert. The figure was reproduced from Hauck et al. [37].

## Data Availability

No new data were created or analyzed in this study. Data sharing is not applicable to this article.

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
