# Peer review of "Vibrational Spectroscopy of Phytochromes"

_biomolecules, 2023, doi:10.3390/biom13061007_

Round 1
Reviewer 1 Report
Dr. Hildebrandt wrote a comprehensive review for vibrational spectroscopy of phytochromes and related proteins. The manuscript is well written and informative, and its publication will merit a field of vibrational spectroscopy as well as biophysics of phytochromes. Actually, this reviewer learned many things from this manuscript. Thus I recommend the publication of this review after the following minor points are considered.
(1) Page 1, line 32
Maybe modify as follow:
.. and PHY (phytochrome-specific) domain (Figure 1A), and ..
(2) Page 1, line 35
Maybe modify as follow:
.. site in the GAF domain (Figure 1D).
(3) Page 2, Figure 1
I suggest the author to include a structure of PVB in Figure 1 or to make a separate figure for PVB. The author may consider a possibility to make panel E of Figure 1 as a separate figure, since this panel is referred at page 6.
(4) Page 2, a figure legend for Figure 1, line 53
Maybe modify as follow:
.. bond network that links biliverdin (BV) to the ..
(5) Page 7
I suggest the author to include a figure that explain syn and anti structures.
(6) Page 11, a figure legend for Figure 3
It is better to explain what left and right panels show.
(7) Page 11, lines 473-479.
This part is redundant to lines 456-463.
(8) Page 12, line 518
Probably a typo: FRSR => FSRS
(9) Page 16, Figure 5
If it is possible, I suggest the author to modify the figure. The numbers in “white or red labels on black areas” are hard to read.
(10) Page 18, line 804
The following sentence is OK? This means that some CBCRs are not found in cyanobacteria.
CBCRs are mainly found in cyanobacteria ..
I guess that the author means
CBCRs are mainly carry phycoviolobilin (PVB) or, in most cases, PCB as the chromophore, attached to a Cys in the GAF domain of the photosensor module.
But it is confusing.
(11) Page 21, line 917
I suggest the author to modify the text something like as follows:
Hirose and Unno group employed UV-vis absorption ..
Author Response
I like to thank the reviewer for the positive comments and the careful reading of the MS. My point-to-point replies are below (in italics).
(1) Page 1, line 32
Maybe modify as follow:
.. and PHY (phytochrome-specific) domain (Figure 1A), and ..
Response: Changed as suggested
(2) Page 1, line 35
Maybe modify as follow:
.. site in the GAF domain (Figure 1D).
Response: Changed as suggested
(3) Page 2, Figure 1
I suggest the author to include a structure of PVB in Figure 1 or to make a separate figure for PVB. The author may consider a possibility to make panel E of Figure 1 as a separate figure, since this panel is referred at page 6.
Response: I prefer not to modify Figure 1 in this respect since PVB is mentioned first in a very late chapter (p. 19). PVB is only relevant in the context of CBCRs. In addition, CBCRs form just a minor part of the review such that I also do not think that an extra figure is justified. At each when PVB is mentioned the appropriate reference are available.
(4) Page 2, a figure legend for Figure 1, line 53
Maybe modify as follow:
.. bond network that links biliverdin (BV) to the ..
Response: Changed as suggested
(5) Page 7
I suggest the author to include a figure that explain syn and anti structures.
Response: The reviewer is right to request further explanation but I do not think that this issue justified an extra figure a long textual explanation. Instead, the reader is now referred to the textbook of Heinz Falk who made outstanding contributions to the chemistry of linear tetrapyrroles. This reference is included in the caption of figure 1, along with a brief explanation.
(6) Page 11, a figure legend for Figure 3
It is better to explain what left and right panels show.
Response: Changed as suggested
(7) Page 11, lines 473-479.
This part is redundant to lines 456-463.
Response: Error was corrected
(8) Page 12, line 518
Probably a typo: FRSR => FSRS
Response: Error was corrected
(9) Page 16, Figure 5
If it is possible, I suggest the author to modify the figure. The numbers in “white or red labels on black areas” are hard to read.
Response: Figure was modified as suggested
(10) Page 18, line 804
The following sentence is OK? This means that some CBCRs are not found in cyanobacteria.
CBCRs are mainly found in cyanobacteria ..
I guess that the author means
CBCRs are mainly carry phycoviolobilin (PVB) or, in most cases, PCB as the chromophore, attached to a Cys in the GAF domain of the photosensor module.
But it is confusing.
Response: The reviewer is right. I corrected this mistake.
(11) Page 21, line 917
I suggest the author to modify the text something like as follows:
Hirose and Unno group employed UV-vis absorption ..
Response: Changed as suggested
Reviewer 2 Report
This is a comprehensive review of the application of vibrational spectroscopic approaches to study phytochromes. The last review on this topic was published in 2015, so many new data appeared since that time to be reviewed. Such kind of a review is necessary for 3 main groups of researchers: (1) ones working with these methods to study phytochromes; (2) ones working with phytochromes by means of other methods to supplement knowledge available thanks to vibrational spectroscopy. With this regard, it is justified to include the article in the special issue; (3) ones working with vibrational spectroscopy searching for its different applications and theoretical methods to analyze their data. As for the last point, it is not the main focus of the review, but plenty of novel works and reviews are cited. I am sure that each of the mentioned groups of readers will be satisfied with the present review. I was thinking about what to add to this review but found nothing relevant to the field that has not been mentioned there. As far as I looked through the References, I found them relevant to the field, there are many new links (27 out of 183 were published in the last 5 years). Certainly, there are 48 self-citations, but it is impossible to avoid them because the author has been fruitfully working in this field for 40 years.To conclude, even thorough reading of the manuscript didn't let me find anything to change.
Author Response
I thank the reviewer for the positive comments and the careful reading of the manuscript.